# Characterisation of the transfer of cluster ions through an Atmospheric Pressure interface Time-of-Flight mass spectrometer with hexapole ion guides

Markus Leiminger[1,2], Stefan Feil[2], Paul Mutschlechner[2], Arttu Ylisirniö[3], Daniel Gunsch[2], Lukas Fischer[1], Alfons Jordan[2], Siegfried Schobesberger[3], Armin Hansel[1,2] and Gerhard Steiner[1,4]

[1]University of Innsbruck, Institute of Ion Physics and Applied Physics, 6020 Innsbruck, Austria
[2]Ionicon Analytik GmbH, 6020 Innsbruck, Austria
[3]University of Eastern Finland, Department of Applied Physics, 70211 Kuopio, Finland
[4]Grimm Aerosol Technik Ainring GmbH & Co. KG, 83404 Ainring, Germany

*Address correspondence to*: G. Steiner (gerhard.steiner@uibk.ac.at) and A. Hansel (armin.hansel@uibk.ac.at)

**Abstract.** Here we present an alternative approach of an Atmospheric-Pressure interface (APi) Time-Of-Flight mass spectrometer for the study of atmospheric ions and cluster ions, the so-called ioniAPi-TOF. The novelty is the use of two hexapoles as ion guides within the APi. In our case, hexapoles can accept and transmit a broad mass range enabling the study of small precursor ions and heavy cluster ions at the same time. Weakly bound cluster ions can easily de-cluster during ion transfer depending on the voltages applied to the ion transfer optics. With the example system of $H_3O^+(H_2O)_{n=0-3}$, we estimate that cluster ions with higher binding energies than 17 kcal/mol can be transferred through the APi without significant fragmentation, which is considerably lower than about 25 kcal/mol estimated from the literature for APi-TOFs with quadrupole ion guides. In contrast to the low fragmenting ion transfer, the hexapoles can be set to a high fragmenting declustering mode for collision-induced dissociation (CID) experiments as well. The ion transmission efficiency over a broad mass range was determined to be in the order of 1%, which is comparable to existing instrumentation. From measurements under well-controlled conditions during the CLOUD experiment, we demonstrate the instrument's performance and present results from an inter-comparison with a quadrupole based APi-TOF.

## 1 Introduction

The study of ion composition in the atmosphere has a long history, and mass spectrometers are being used as an important tool in elucidating their identity and concentrations since the early days. Galactic cosmic rays (GCR) are the main ionisation source in the atmosphere, while radioactive decay (of radon) is more relevant at ground level. Minor entries originate from lighting, power lines and combustion sources (Curtius, 2006). Higher ion number concentrations are detected in the upper atmosphere and lower number concentrations at ground level. Typically, up to ten thousand ions per $cm^3$ can be observed within the troposphere having a life time of a few hundred seconds (Ferguson and Arnold, 1981; Hirsikko et al., 2011). Despite their low

abundance, ions can play an important role in atmospheric new particle formation via ion-ion-recombination and ion-induced
nucleation (Kirkby et al., 2016) as well as in atmospheric electricity.
In the 1970's, F. Arnold and co-workers were the first to study the composition of ions in the lower stratosphere and upper
troposphere. In the positive ion spectrum, detected signals were mainly attributed to hydrated hydronium clusters
$H_3O^+(H_2O)_{n=1-4}$ and protonated organic vapours (Arnold et al., 1977, 1978). For negative ions, clusters of de-protonated acids
like $NO_3^-(HNO_3)_m$ and $HSO_4^-(H_2SO_4)_p(HNO_3)_s$ were identified in the mass range of 1 to 280 amu (Viggiano and Arnold,
1981). At ground level, the composition of the main tropospheric ions was also studied by F. Eisele and co-workers with a
quadrupole mass spectrometer (Eisele, 1986; Perkins and Eisele, 1984). Using collision-induced dissociation (CID), they
identified the 'core' ions of hydrated clusters showing that positive core ions consist mainly of protonated amines. The latter
were examined using Tandem-mass spectrometry which helped to identify pyridine and its homologues  (Eisele, 1988). Back
then, F. Eisele already observed a manifold of tropospheric ions up to 700 amu in the positive ion mass spectrum, but the low
mass resolving power of the quadrupole mass analyser was a bottleneck for revealing their sum formula. Tandem mass
spectrometry was not performed for these heavy ions for several reasons like insufficient sensitivity and the natural variability
(Eisele and Tanner, 1990).
The development of an Atmospheric-Pressure interface Time-Of-Flight mass spectrometer (APi-TOF MS, Aerodyne Research
Inc. and Tofwerk AG) overcame the limitations of quadrupole mass analysers regarding mass resolving power, duty cycle and
mass range. Junninen et al. (2010) demonstrated that this instrument is suitable to detect many unknown ions in the atmosphere
and assign sum formulas to many mass peaks for the first time (Ehn et al., 2010; Junninen et al., 2010). Especially in the field
of atmospheric new particle formation, the APi-TOF enabled the study of ion formation starting from single molecules such
as sulphuric acid, ammonia, amines and highly oxygenated organic molecules (HOM) to the formation of molecular clusters
of sizes with a mobility equivalent diameter of 1-2 nm (Almeida et al., 2013; Kirkby et al., 2016; Kürten et al., 2014;
Schobesberger et al., 2013). In the last couple of years, the APi-TOF was the key instrument for many scientific studies of new
particle formation in both laboratory and field settings (Bianchi et al., 2016; Kirkby et al., 2011; Sipilä et al., 2016).
However, questions arose about fragmentation of cluster ions inside the APi-TOF instrument during the ion transfer from
ambient pressure through the two quadrupoles and the following lens system to the detector (Ehrhart et al., 2016). It remained
unclear if additional ligands besides water molecules might be lost during the ion transfer as well. In a recent publication by
Olenius et al. (2013), the authors concluded that fragmentation might be a reasonable explanation for the observed difference
in measured and modelled cluster ion distributions of $HSO_4^-(H_2SO_4)_m(NH_3)_n$ clusters (Olenius et al., 2013).
Few publications explicitly studied fragmentation inside the APi-TOF mass spectrometer (Bertram et al., 2011; Brophy and
Farmer, 2016; Lopez-Hilfiker et al., 2016). Bertram et al. (2011) showed that fragmentation of cluster ions is strongly sensitive
to the voltage settings in the APi. Lopez-Hilfiker et al. (2016) as well as Brophy and Farmer (2016) used two different types
of Chemical Ionisation (CI-) APi-TOF to study fragmentation of reagent-adduct-cluster ions. Both found that the electric field
inside the APi could be tuned to a low fragmenting "clustered" setting and a high fragmenting "declustering" setting. Even
using the low fragmenting setting, however,  the transfer of weakly bound cluster ions was evidently affected by fragmentation
inside the APi (Brophy and Farmer, 2016; Lopez-Hilfiker et al., 2016). Here, the question arises which cluster bond strengths
are how strongly affected by fragmentation.
For the instrument configuration used by Lopez-Hilfiker et al. (2016), Iyer et al. (2016) found that for iodide-($I^-$)-chemical
ionisation, adduct-molecule clusters with binding energies above 25 kcal/mol are mostly detected with maximum sensitivity
at the collisional limit by comparing experimentally measured sensitivities with modelled binding energies. Cluster ions below
this threshold suffer from lower sensitivities, likely, due to non-thermal dissociation during the ion transfer inside the mass
spectrometer (i.e. partial fragmentation). It remains unclear, if this threshold can be explained by fragmentation in the APi or
by the loss of weakly bound ligands during the charging process of a neutral cluster by the reagent ion in the Ion-Molecule-
Region (IMR) of the ToF-CIMS (Kurten et al., 2011). In the supplement, however, the authors conclude that fragmentation in
the APi of their ToF-CIMS is more reasonable. Furthermore, they also state that cluster ions with binding energies below 10
kcal/mol may not be detectable at all (Iyer et al., 2016). Consequently, there may be two threshold binding energies, one below
which partial fragmentation of cluster ions can be expected and the other one below which the non-detection of cluster ions is
almost certain. Quantifying these thresholds (e.g. around 10 and 25 kcal/mol for the APi configuration in Iyer et al., 2016) can
help characterising the ion transfer of APi-TOF instruments.
In those previous studies, the APi's declustering strength was deliberately manipulated by varying the electric potential
gradients between two ion optic parts in the APi, e.g. between the skimmer and the second quadrupole (Brophy and Farmer,
2016; Lopez-Hilfiker et al., 2016). This electric field is located at the transition from the first to the second pressure stage
where the gas pressure drops from two hundred to a few Pa. The cluster ions accelerated by the electrical field can therefore
attain relatively high energies via collisions (Zapadinsky et al., 2019, at > 100 Pa, collisions tend to be too frequent and hence
low in collision energy, at << 1 Pa, collisions tend to be too rare due to the increased mean free path). Hence, the transition
region from the first pressure stage to the second one is also a transition from multi to single collision conditions.
The role of the quadrupoles in the fragmentation of cluster ions has not been investigated so far. From theory, there are some
differences with regard to the ion transfer properties comparing the quadrupole to higher order multipoles that can mainly be
explained by the number of rods. To radially trap or guide ions of various mass-to-charge (m/z) ratios through a multipole a
radiofrequency (RF) with amplitude $V_0$ is applied on alternating rods. Ions of low m/z are efficiently trapped with higher
frequencies and lower amplitudes while ions of high m/z can be more efficiently transferred with lower frequencies and higher
amplitudes. The time-averaged radial trapping field within a multipole of 2n electrodes can be described with the effective
potential $V^*$ (Gerlich, 1992):

$$V^* = \frac{n^2}{4} \frac{q^2}{m\Omega^2} \frac{V_0^2}{r_0^2} \left(\frac{r}{r_0}\right)^{2n-2} \tag{1}$$


Here, we have the charge q, the ion mass m, the angular frequency $\Omega$, the amplitude $V_0$, the inner radius of the electrode
arrangement $r_0$ and the radial distance of the ion r inside the multipole. In general, the effective potential $V^*$ is high close to
the rods and low close to the centre. The slope between multipole rods and its centre depends on the rod number, see Fig. S0.
A higher rod number further provides a more homogenous trapping field. The trapping fields of RF-only multipoles do not
affect the axial kinetic energy of ions, but can affect the radial ion energy (Armentrout, 2000).
From Equ. 1, it can be seen that the effective potential varies with $(r/r_0)^{2n-2}$. A quadrupole (n=2) has a quadratic dependence
$(r/r_0)^2$ while a hexapole depends on $(r/r_0)^4$. Consequently, the effective potential of a quadrupole increases much closer to the
centre of the ion guide compared to a hexapole. On the one hand, this results in an efficient focusing of the ions for a
quadrupole, but on the other hand, this yields strong perturbations of ions in radial direction and thus, the ion kinetic energies
are not well defined. Here, a hexapole has a much lower impact on the radial energy due to a larger field free region, as the
effective potential is flatter close to the centre and higher close to the rods. Compared to higher order multipoles that have an
even larger field free region, a hexapole still offers a more pronounced focusing power.
The $n^2$-dependence of the effective potential further means that for the same RF settings, a hexapole has a stronger trapping
field over a quadrupole of a factor of 9/4. To transfer ions of high m/z with the same efficiency, a quadrupole would require
higher RF settings which in turn would lead to an increased effective potential not only close to the rods but also in the centre
according to the $(r/r_0)^2$ dependence of the effective potential. From this, higher order multipoles should in general show a lower
impact on the stability of cluster ions.
Further, it is important to mention the mass discrimination properties of multipole ion guides (Heinritzi et al., 2016). Small
ions can be lost due to unstable trajectories at higher RF settings on the multipole, which is known as the low mass cut-off.
However, heavy ions typically need a stronger effective potential within the ion guide to be efficiently focused and transferred
(see Equ. 1). Therefore, the efficient transmission of small and heavy ions in a multipole ion guide depends on the mass
window of the multipole. Higher order multipoles are recommended for the transfer of a broader mass window ranging from
low to high masses (Gerlich, 2004). For quadrupole ion guides, the effect of a dramatic cut-off at low masses is especially
pronounced due to the much narrower field free region within the quadrupole compared to higher order multipoles (Gerlich,
1992). In general, multipoles can be tuned to a mass window of interest. In the field of atmospheric new particle formation, a
broad mass range is essential to get a complete understanding of the nucleating ions. Primary ions like $NO^+$, $O_2^+$ or $H_3O^+$ have
different charging properties and the detection of small ions can therefore help to identify the composition of heavy cluster
ions by revealing likely ionisation pathways. Thus, information could be lost due to mass discrimination effects for small and
heavy ions, respectively. Here, hexapole ion guides show advantageous properties regarding the ion transfer.
Ion trajectory simulations through a quadrupole, a hexapole and an octopole by Hägg and Szabo (1986) showed that higher
multipoles (n≥3) are more suited for guiding ions while only the quadrupole can be used as a mass analyser. In an
accompanying study, the authors found that the transmission through higher multipoles depends on the initial conditions of
the ion beam, e.g. initial position or velocity (Hägg and Szabo, 1986). The reason for this is that the x- and y-coordinates are
no longer independent compared to the quadrupole. An overall lower transmission efficiency of ions could be a likely
consequence. Hägg and Szabo (1986) found in another study that the multipoles of even-order like the octopole have more
stable trajectories because the opposing electrodes have the same sign whereas multipoles of odd-order like the hexapole have
opposing electrodes of opposite sign (Hägg and Szabo, 1986). This would be one benefit using an octopole over a hexapole
despite otherwise similar transfer properties. For further ion trajectory simulations in multipole ion guides with focus on
phenomena like collisional cooling and radial stratification of different m/z ions due to ion-ion and ion-neutral interactions we
refer the reader to Tolmachev et al. (2003) and references therein. The main properties of multipole ion guides are summarised
in Table 1.
In the present study, we introduce the ioniAPi-TOF with hexapole ion guides. We characterise the performance of the ioniAPi-
TOF regarding ion transmission efficiency, mass range and the impact of electric fields in fragmenting cluster ions.
Additionally, we present an inter-comparison with a state-of-the-art quadrupole based APi-TOF during the CERN CLOUD
experiment performed in fall 2017 and discuss similarities as well as differences in the instrument's performance.

## 2 Instrument and methods

### 2.1 The ioniAPi-TOF

The ioniAPi-TOF mass spectrometer consists of a laminar flow inlet, an Atmospheric-Pressure interface (APi) including two
hexapole ion guides, an ion transfer optic and an orthogonal extraction, reflectron Time-Of-Flight (TOF) mass analyser (see
Fig. 1).
The laminar flow inlet draws atmospheric ions from the ambient to the inlet of the mass spectrometer via an adjustable flow
of 1 to 15 L/min. The inlet is made of a stainless-steel tube with a length of 10 cm and a diameter of ½" inch. Within this tube,
a core-sampling probe is placed in front of the entrance aperture of the ioniAPi-TOF with an inner diameter of 2.5 mm and a
length of 25 mm as indicated in Fig. 1.
The entrance aperture has a diameter of 0.4 mm yielding an inlet flow of 1.1 L/min from ambient pressure into the mass
spectrometer. Skimmers with bore diameters of 1.2 mm separate the different pressure stages. Two hexapole ion guides of the
same length are installed in the first and second pressure stage of the Atmospheric-Pressure interface. The first hexapole is
running at a frequency of 1 MHz and an amplitude of 200 $V_{pp}$. The frequency of the second hexapole is about 5.5 MHz with
an amplitude of 600 $V_{pp}$. The third pressure stage contains an ion transfer optical lens system consisting of two lens stacks. It
focuses the ion beam coming from the hexapoles and transfers it to the orthogonal extraction region of the mass spectrometer.
As Time-Of-Flight mass analyser, we chose the ioniTOF1000 platform of IONICON Analytik GmbH (Innsbruck, Austria). It
is a compact Time-Of-Flight mass spectrometer with a short ion flight path of roughly 0.5 m and therefore expected to have a
sufficiently high ion transmission efficiency, which is important due to the low abundance of atmospheric ions. The same
Time-of-Flight mass analyser was already presented in Müller et al., (2014). The ioniTOF1000 is made of a multistage
orthogonal-extraction region consisting of a pusher and four mesh electrodes supplied with a reference, pull, grid as well as a
drift tube cage voltage.
Depending on the desired mass range of interest, the extraction frequency can be adjusted to measure ions of a mass-to-charge
(m/z) ratio of up to 10,000 Th (1 Thomson = 1 Da/e$^-$). For the results presented herein, the extraction frequency is typically set
to 30 kHz to measure ions up to 2,000 Th.
A double-stage reflectron is used for an improved mass resolution leading to a V-shaped ion flight path. Ions are post-
accelerated and detected with a multichannel plate (MCP) stack with a voltage of ~2200 V. Compared to Müller et al. (2014)
we achieved a mass resolution at full-width half maximum (FWHM) of ~2000 for ions above m/z 100, see Fig. 2.
A Hyco 4-cylinder diaphragm pump is used at ambient pressure to draw air through the laminar flow inlet. A Pfeiffer Vacuum
ACP 40 roots pump is used as fore pressure pump to reduce the pressure in the cell of the first hexapole in the ioniAPi-TOF
to 2.3 mbar. Three Pfeiffer Vacuum HiPace 80 turbo-molecular pumps, which are connected to a MD1 diaphragm pump of
Vacuubrand, evacuate the ion transfer optics region and the TOF mass analyser. Together they maintain a typical pressure of
a few $10^{-3}$ mbar in the second hexapole of the ioniAPi, followed by $10^{-4}$ mbar within the lens stacks and a few $10^{-6}$ mbar in the
TOF mass analyser.
As described in Müller et al. (2014) a time-to-digital converter (TDC) is used to convert the MCP signals into ion counts per
time-bin. The applied extraction frequency of 30 kHz results in about 280,000-time bins. The IONICON TOF 3.0 software is
used for data acquisition. The data is stored in the HDF5-file format (HDF5-group). The ioniAPi-TOF allows detection of ions
in positive and negative ion mode. In this work, we present results of the positive ion mode only.
**2.2 Cluster Calibration Unit**
The Cluster Calibration Unit (CCU) allows the calibration of the mass-axis over a broad mass range and of the mass-dependent
transmission efficiency of (CI-)APi-TOF mass spectrometers (Heinritzi et al., 2016). For this purpose, the CCU consists of an
electrospray ionisation source (ESI), a "Vienna"-type high-resolution Differential Mobility Analyser (UDMA, Steiner et al.,
2010) and a Faraday-Cup electrometer (FCE, Winklmayr et al., 1991); see Fig. 3.
Millimolar solutions of tetra-alkyl-ammonium halides dissolved in acetonitrile are used for the ESI, see Table 2 for details
(Ude and Fernández de la Mora, 2005). By applying a high voltage of a few kV, the ESI generates cluster ions of the desired
polarity. A transport flow of typical 14 L/min transfers the ions over a distance of a few centimetres from the ESI directly into
the UDMA. Within the UDMA, the ions are classified in terms of their electrical mobility. In this study, a filtered recirculating
sheath flow of about 700 L/min was used. Under these conditions and the given geometry, the resolving power of the UDMA
was around 10 to 15, which is sufficient to distinguish ionic monomers, dimers and trimers of one selected calibration
compound within the ion mobility spectrum. An exemplary ion mobility spectrum is shown in the supplement Fig. S1. For
details on the definition on the resolving power of DMAs we refer to Flagan, 1998.
To retrieve the transmission at a desired mass-to-charge (m/z) ratio in the mass spectrometer, the UDMA can be set to a
constant voltage that corresponds to a specific electrical mobility. Consequently, only ions of the corresponding electrical
mobility will pass the UDMA. The aerosol flow coming from the outlet of the UDMA is guided through an 8 mm stainless
steel tube of 10 cm length. After this length, the flow is separated via a Y-shaped flow splitter with an angle of 40° between
the two outlet tubes to reduce inhomogeneities of the sample flow. For the same reason, the flow rates to both the FCE and the
ioniAPi-TOF are set equally to 6 L/min resulting in an overflow of 2 L/min.

## 2.3 Experiments with a corona ion source

For the comparison of a low fragmenting (clustered) setting (voltage difference: dV = -1.4 V) and a high fragmenting
(declustering) setting (voltage difference: dV = -10.0 V) described in chapter 3.2, ions were generated with a corona ion source.
A ½" T-piece was connected to the ½" laminar flow inlet in front of the ioniAPi-TOF. Cleaned and dried lab air was drawn in
a straight line through the T-piece. The corona needle was placed through a ½" plug into the T-piece orthogonally to the flow
direction. The needle tip was sitting below the air flow.
A voltage of +1.7 kV lead to the ionisation of ambient air streaming into the direction of the entrance aperture of the mass
spectrometer. Hereby, a large variety of ions was produced via ion-molecule reactions and charge transfer covering a mass
range from 18 to 1000 Th. The reaction time within the laminar flow inlet was approximately 35 ms. In the course of the
experiment, high amounts of $H_3O^+(H_2O)_n$ were needed to study the fragmentation of these cluster ions. The corona ion source
yielded sufficient and stable ion signals for constant flow conditions inside the inlet tube as will be shown in chapter 3.2, Fig.

210 7.

## 2.4 CLOUD experiment

To test the performance of the ioniAPi-TOF under the high-demanding conditions of a long-term measurement campaign with
the challenge of various experimental conditions and different chemical systems, we participated in the CLOUD 12 campaign
in fall 2017. The CLOUD (Cosmics Leaving OUtdoor Droplets) experiment at the European Centre for Nuclear Research
(CERN) studies the influence of galactic cosmic rays (GCR) on atmospheric new particle formation under very well-controlled
conditions (Duplissy et al., 2016; Kirkby et al., 2011). This effect can be studied by comparing the experiments at ground level
GCR ion pair production rates to experiments under neutral conditions inside the chamber where a high voltage field is turned
on. Upper tropospheric ion pair production rates and ion number concentrations can be realised via a π-beam of 3.5 GeV/c
from the CERN Proton Synchrotron. The chamber is made of electro-polished stainless steel with a volume of 26.1 m³. At the
top and at the bottom of the chamber, two fans made of stainless steel are used for homogeneous mixing of the air yielding
mixing times of a few minutes. To study a wide range of tropospheric conditions, a thermal housing allows experiments at
temperatures ranging from 203 to 310 K with a stability of 0.1 K. A very clean atmosphere is obtained using cryogenic $N_2$ and
$O_2$ in the natural ratio of 79:21 with a level of contaminant vapours in the sub-$ppt_v$ range (Schnitzhofer et al., 2014). The effect
of relative humidity can be studied by adjusting the flow rate of ultrapure de-ionized water being vaporised into the chamber.
Ozone is produced via UV photolysis of $O_2$. The volume-mixing ratio of $O_3$ can be controlled by the flow rate. Further trace
gases like $SO_2$, $NH_3$, isoprene ($C_5H_8$) or α-pinene ($C_{10}H_{16}$) can be introduced separately via a gas handling system.
During the measurements at the CLOUD experiments, we used a critical orifice at the exhaust of the inlet to maintain a constant
flow rate of 12.6 L/min as this was found to be the optimal setting regarding the total ion signal intensity. The ioniAPi-TOF
inlet line was connected via a flow splitter with the PTR3 (Breitenlechner et al., 2017) to the same CLOUD sampling port.
Due to reasons of limited space around the CLOUD chamber, the ioniAPi-TOF was mounted on top of the PTR3.
Consequently, it could not be connected via a straight line to the flow splitter. We connected the instrument with two 30 cm
long flexible well tubes and one additional straight tube. All tubes were made of stainless steel. In total, the sigmoidal-shaped
inlet line to the flow splitter had a length of 1.2 m. Together with the length of the sampling probe that reached into the
chamber; the total length of the inlet line was about 1.95 m and had a diameter of ½". Besides wall losses due to the length of
the inlet line, the flexible well tubes might have resulted in an additional loss factor due to their rippled inner surface.
**2.5 The APi-TOF**
The operation principle of the APi-TOF of the University of Eastern Finland (UEF) is similar to what has been extensively
reported in previous publications (Junninen et al., 2010; Schobesberger et al., 2013). The instrument was directly connected
to the CLOUD chamber through a 30 cm long stainless-steel tubing with an outer diameter of 1" (25.4 mm), which was then
reduced to 10 mm diameter in the last 10 cm of the tubing. The flow rate inside the sampling tube was in total 9 L/min all the
way to the 0.3 mm diameter sampling pinhole of the instrument. From the 9 L/min total flow, 0.8 L/min entered the instrument.
The UEF APi-TOF was operated in positive ion mode for the experiments shown here with ion guiding quadrupoles operating
at pre-defined "high mass" settings having a mass range of about 100 – 2000 m/z.
The main differences in the configuration of both instruments are listed in Table 3. The configuration of the ion transfer system
in the APi shows major differences due to the use of segmented quadrupoles in the UEF APi-TOF while  non-segmented
hexapoles are used in the ioniAPi-TOF as well as other geometric factors like e.g. skimmer orifice diameters and distances.
The different lengths of the TOF mass analysers explain differences in the mass resolving power and the extraction frequencies.
**2.5 Data analysis and post-processing**
The data of the UEF APi-TOF were processed using the MatLab based tofTools package Version 6.11 (Junninen et al., 2010).
We used the Ionicon PTR-MS Viewer 3.2 and TOF data processing scripts written by Lukas Fischer for data analysis of the
UIBK ioniAPi-TOF (for TOF data processing scripts see Breitenlechner et al., 2017).
**3 Results**
**3.1 Characterisation of the transmission efficiency**
The overall absolute transmission efficiency of the ioniAPi-TOF was determined with the Cluster Calibration Unit for a mass
range of 74 to 1640 Th. The transmission efficiency of a selected m/z was determined by the ratio of ion count rates measured
with the ioniAPi-TOF and the FCE. The substances listed in Table 2 were used as calibration standards. Monomer, dimer,
trimer and tetramer cluster ions were produced with the ESI and selected each as a monodisperse aerosol via the UDMA. The
smallest ion was the monomer of tetra-methyl-ammonium iodide at m/z 74 and the heaviest cluster ion used was the tetramer
of the ionic liquid with m/z 1640.
The mass spectrum of a monodisperse aerosol typically has major counts at the m/z peak of the mobility selected ion. Minor
counts of ions of m/z < 100 like $O_2^+$, $H_3O^+(H_2O)_{n=0-3}$, $NH_4^+(H_2O)_{n=0-2}$ as well as protonated acetonitrile clusters $H^+(C_2H_3N)_{1-2}$
were also observed. Additionally, minor peaks of impurities or fragments were observed, see Fig. S2. In the case of dimers,
their signal still showed the highest intensity. In addition, a peak at the m/z of the monomer appeared with a relative abundance
of less than 10 %. The observation of fragments was even more pronounced in the case of trimers. Here, the count rates of
monomers and dimers reached in some cases similar intensities compared to the trimer, although only the trimer was expected.
This was not only observed for all mobility standards in Table 2, but also in different types of APi-TOF mass spectrometers
using the CCU. We could observe the same fragmentation pattern with the ioniAPi-TOF as well as with the UEF APi-TOF in
the course of the inter-comparison, and with an H-TOF, Tofwerk AG Thun Switzerland, without an APi interface in the
laboratory at the University of Innsbruck (UIBK). This H-TOF was not equipped with a typical APi as it is part of a PTR-SRI-
TOF MS (Graus et al., 2010). For the experiments with the UDMA, we mounted a simple single pressure stage. This single
pressure stage consisted of a front plate with a critical orifice diameter of 0.3 mm and two electrode lenses that were connected
to the sampler plate of the H-TOF. A pressure of 2 mbar in the single pressure stage was achieved with a pre-pressure pump.
To our knowledge, there exist only a few detailed reports of observations of such fragmentation patterns for the standards we
used here. Heinritzi et al. (2016) reported fragmentation of iodide dimers in the negative ion mode. While Junninen et al.
(2010) did not observe such fragmentation with the calibration standards, only at a mobility diameter of 1.6 nm a fragment
possibly due to an impurity was reported.
The aforementioned observation can either be interpreted as fragments or as the result of a broad tail of the UDMA's transfer
function allowing ions of high abundance to be still partially transferred despite not having the expected ion mobility. Further,
also multiply charged ions with the same ion mobility could pass the UDMA and evaporate or fragment afterwards leading to
the formation of monomers and dimers which are then detected (Rus et al., 2010). No peaks of multiply charged ions were
observed in the mass spectrum. So far, we exclude fragmentation inside the ioniAPi as explanation, as we will show in section
3.2. that, when using the low fragmenting setting in the ioniAPi (voltage difference: -1.4 V) the cluster ion  $H_3O^+(H_2O)_3$ does
not appear to fragment almost at all, even though it is relatively weakly bound (binding energy: $BE(H_3O^+(H_2O)_3) = -\Delta H = 17$
kcal/mol, (Meot-Ner, 1984)). For much heavier cluster ions, such as produced with the calibration standards, even higher
collision energies would be necessary for a fragmentation of the observed intensity. The relationship between the energy in
the lab system $E_{lab}$ and the centre-of-mass energy $E_{CM}$ is shown in Equ. 2 (Armentrout, 2002). Here, m is the mass of the buffer
gas (air) and $m_{ion}$ the mass of the ion. $E_{CM}$ is proportional to the reciprocal of the ions' mass $m_{ion}$. With increasing mass, higher
electric fields would be necessary to reach sufficient collision energies for heavy ions.

$$E_{CM} = \frac{m}{m+m_{ion}} \cdot E_{lab} \tag{2}$$

The conversion into the centre-of-mass frame of reference allows the estimation that for ions with a high m/z, e.g. m/z>250
Th, the collision energy under low fragmenting settings and air molecules as buffer gas should not be sufficient to explain the
observed peak pattern by fragmentation.
However, the fractions of fragment signals can be corrected as done in Heinritzi et al. (2016). For this purpose, we assume that
the fragmentation occurs outside the APi. Thus, the electrometer counts the fragments as well. In general, ions of different m/z
have different transmission efficiencies through an APi. To obtain the transmission of the monomer, solely the sum of count
rates at the monomer mass and its isotope peaks was divided by the expected count rate that was determined from the current
measured with the electrometer. With the obtained monomer transmission efficiency, the electrometer signal was corrected to
determine the transmission for the dimer. Further, the transmission factor of the monomer and the corrected transmission factor
of the dimer were used to determine the transmission of the trimer.
In the end, this leads to the overall absolute transmission efficiency shown in Fig. 4. An overall transmission efficiency of
about 1 % was found. Considering the instruments background noise, this corresponds to a detection limit of roughly $5 \cdot 10^{-3}$
ions/cm$^3$ for 5-min integration time and $5 \cdot 10^{-4}$ ions/cm$^3$ for one-hour integration time. The error in determining the transmission
efficiency due to fragment peaks was found to be less than 10 %. In general, the transmission is highest in the mass range from
200 to 600 Th and decreases for heavier ions. The transmission of small ions was only determined in the course of one
experiment where it seems to decrease sharply to values as low as for heavy ions. Nevertheless, we later observed the highest
individual ion count rates are under standard (low fragmenting) settings highest at ions below m/z 100 (for example see Fig.
9). This may indicate that small natural ions are more than one order of magnitude more abundant than heavier ions or that the
transmission at m/z 74 is underestimated.
The transmission efficiency was determined for both the low fragmenting (voltage difference: dV = -1.4 V) and the high
fragmenting (voltage difference: dV = -10.0 V) setting for comparison. As shown in Fig. 4, the low-fragmenting setting yields
a higher transmission efficiency for most of the mass range. Despite an overall lower transmission, the high-fragmenting setting
offers a slightly higher transmission for heavier ions, here m/z 1391, due to the better focusing of heavier ions. This resembles
a shift or a tilting of the transferred mass window. Overall, though, both settings offer a comparable high ion transmission.
The data points (Fig. 4) determined after the CLOUD campaign for the low fragmenting setting are comparable to the
calibration done in the beginning of the campaign.
**3.2 Characterisation of the ion transfer**
In the following, we address the question of fragmentation inside the ioniAPi-TOF. As mentioned in the introduction, recent
studies demonstrated that the electric potential difference between parts of the ion optics in the APi can be used to study
collision induced fragmentation of cluster ions, e.g. a voltage difference dV between the skimmer and the second multipole
(Brophy and Farmer, 2016; Lopez-Hilfiker et al., 2016).
Further, it was shown for the APi-TOF that fragmentation of clusters is most likely for pressures between 1.0 and 0.01 mbar
and elevated electric fields (Zapadinsky et al., 2019). Similar conditions can also be found and set in the ioniAPi-TOF. With
regard to the conditions presented in Zapadinsky et al. (2019), the most critical region in the ioniAPi-TOF would be the
transition from the first to the second pressure stage. In the current instrument configuration, it was not possible to apply any
other electric fields to the first pressure stage aside from the RF frequency and amplitude. Therefore, we exclude fragmentation
due to axial electric fields in this region. Downstream of the second pressure stage, ions can be accelerated to even higher
energies compared to previous regions. However, at a pressure below $10^{-4}$ mbar the number of collisions is too low due to a
mean free path of above 50 cm. In the present study, we focus on the second pressure stage where fragmentation is most likely.

### 3.2.1 Low and high fragmenting setting

To compare the afore-mentioned low and high fragmenting settings, hydrated hydronium clusters ($H_3O^+(H_2O)_{n=0-3}$) were used
as a model system due to their well-known binding energies, see Table 4 (Meot-Ner, 1984). In Fig. 5, the distribution of the
hydrated hydronium clusters is exemplarily shown for both settings. Ions were produced using a corona ion source in front of
the inlet as introduced in chapter 2.3. We used the fits of the ion transmission efficiency of the low and the high fragmenting
setting from Fig. 4 to correct the individual ion intensities for all the water-clusters with respect to transmission effects.
In the low fragmenting setting, the higher order and weakly bound hydrated hydronium clusters $H_3O^+(H_2O)_2$ and $H_3O^+(H_2O)_3$
show the highest abundance (Fig. 5). The high fragmenting setting (highest dV) overall leads to the cluster distribution shifting
to smaller and more strongly bound hydrated hydronium clusters, $H_3O^+(H_2O)_3$ largely dissociates, thereby reducing its intensity
by a factor of 10.
The peak at m/z 91.06, assumed to be $H_3O^+(H_2O)_4$, is also included in the figure. Its signal intensity seems to behave as
expected for the low and the high fragmenting setting because it shows a reduction for the latter setting.
The intensity of $H_3O^+$ increased by a factor of 25 for the high fragmenting setting. Although this is a significant increase, the
new cluster equilibrium ends with $H_3O^+(H_2O)$ and $H_3O^+(H_2O)_2$ showing the highest intensities. Evidently, a voltage difference
of -10 V which was the maximum adjustable voltage setting is not enough to completely fragment $H_3O^+(H_2O)$ cluster ions
(bound most strongly, $-\Delta H = 31.5$ kcal/mol; Table 4).

### 3.2.2 Declustering scan

A so-called declustering scan investigates the relation of voltage settings in the APi to the binding energy of cluster ions
(Lopez-Hilfiker et al., 2016). In the current configuration of the ioniAPi-TOF, no axial electric fields can be applied to any
parts of the first pressure stage as explained previously. Therefore, the dV scan is obtained in a slightly different way compared
to the one described in Lopez-Hilfiker et al. (2016). In Lopez-Hilfiker et al. (2016), the whole first pressure stage is shifted
towards a more negative dV while the voltages downstream remain constant. In our case, no shift of the first pressure stage is
currently possible. Therefore, the first pressure stage remains at zero potential. The voltage difference between skimmer-1 and
the second hexapole was stepwise increased, here by reducing the DC offset of the hexapole (Fig. 6). The ion optics following
the second hexapole were set to one setting during the declustering scan to maintain a high transmission efficiency with a

constant voltage of -13 V at the first lens that follows the second hexapole. The declustering scan started from dV = 0 V to -10 V in steps of 1 V, skimmer-1 being grounded. Ions were generated with a corona ion source as before.

Fig. 7 shows the dV scan for four hydronium cluster ions $H_3O^+(H_2O)_{n=0-3}$. The initial cluster distribution may look differently depending on the conditions in the first pressure stage like pressure or electric fields, e.g. different RF settings on the first hexapole can alter the mass dependent transmission. The count rates of each ion are normalised to its initial count rate during the scan. Increasing the dV from 0 to -3 V increases the transmission of all four clusters.

Each increase in dV results in a higher collision energy. This explains why primarily the higher order hydrated hydronium clusters show a decrease for the lowest voltage steps. First, $H_3O^+(H_2O)_3$ is collisionally fragmenting due to its low binding energy ($-\Delta H$ = 17 kcal/mol, see Table 4). In the centre-of-mass system, the collision energy needed to break the cluster bond corresponds to the Gibbs free energy of the $H_3O^+(H_2O)_3$ cluster ($\Delta G$ = -9 kcal/mol at 298 K, see Table 4). Although the Gibbs free energy is more accurate in describing the energy of a cluster ion within this process, the estimation of the Gibbs free energy is not straightforward. This is due to the uncertainty of temperature in the transition from the first to the second pressure stage. Therefore, we exemplarily determined the $\Delta G$-values for the hydrated hydronium clusters at a temperature of 298 K in Table 4. In the following, we use the binding energy ($-\Delta H$).

Further increase of dV results in the fragmentation of $H_3O^+(H_2O)_2$ which has a slightly higher binding energy ($-\Delta H$ = 20 kcal/mol). While larger clusters are fragmenting an increase is observed for $H_3O^+(H_2O)$. Above a dV of -8 to -9 V, also the intensity of $H_3O^+(H_2O)$ starts showing a decrease. Here, the collision energy is already high enough to partially fragment $H_3O^+(H_2O)$ that has a much higher binding energy ($-\Delta H$ = 31.5 kcal/mol). $H_3O^+$ shows a steady increase which is pronounced for higher dV. $H_3O^+(H_2O)_2$ shows no significant response to the decrease of the $H_3O^+(H_2O)_3$ ion. This can be attributed to an overall low count rate of $H_3O^+(H_2O)_3$ and a much higher count rate of $H_3O^+(H_2O)_2$. Fragmentation of $H_3O^+(H_2O)_3$ will therefore not significantly increase the $H_3O^+(H_2O)_2$ count rate.

For such a dV scanning procedure, Lopez-Hilfiker et al. (2016) found a linear relationship between the voltage corresponding to the half signal maximum of a cluster, the so-called $dV_{50}$, and the binding energy (Lopez-Hilfiker et al., 2016). In accordance to that study, we used a non-linear least square sigmoidal model to fit the data points. From the fit, we determined a $dV_{50}$ of -5.4 and -7.5 V for $H_3O^+(H_2O)_3$ and $H_3O^+(H_2O)_2$, respectively. The higher $dV_{50}$ obtained for $H_3O^+(H_2O)_2$ is consistent with its binding energy being higher than the one of $H_3O^+(H_2O)_3$ (see Table 4).

As the voltage at the first lens is set to -13 V, fragmentation between the second hexapole and the following lens might dominate the first few voltage steps. To distinguish the role of the region from the skimmer to the entrance of the second hexapole and the region from the exit of the second hexapole to the following lens, we show additional experiments in the supplement. With a high-resolution ioniAPi-TOF, we conducted the same experiments and show with Fig. S4 that both instruments show a good agreement in the responses of the hydronium ion distribution to the dV scanning procedure used in Fig. 7. Fig. S5 shows a declustering scan between skimmer-1 and the second hexapole. Here, the second hexapole and the following lens are stepped synchronously with a constant voltage difference of -1 V between both ion optic parts. This is necessary to maintain sufficient transmission. From Fig. S6, it can be concluded that this small offset should not affect the

fragmentation. The dV scan in Fig. S5 shows that the $dV_{50}$ is shifted to lower values, see Table 4. This shows that the $dV_{50}$
values in Fig. 7 are offset by the voltage at lens-1. In Fig. S6, a declustering scan between the exit of the second hexapole and
the following first lens with lens-1 and lens-2 being stepped synchronously reveals that a dV below -9 V mainly increases the
ion transmission. Only above -9 V the voltage difference from lens-1 to the exit of the second hexapole is high enough to
induce fragmentation of the weaker bound hydronium cluster ions. Fig. S7 shows that a voltage scan between lens-1 and lens-
2 has no effect on the hydronium cluster ion distribution. To conclude, the region between skimmer-1 and the second hexapole
is the region where cluster ions are most likely affected by fragmentation depending on the voltage settings compared to the
other probed regions.
A potential source of uncertainty on the experiments with hydronium cluster ions may be the fragmentation of larger hydrated
hydronium clusters $H_3O^+(H_2O)_n$ with n>3. Such clusters could potentially form on collisions with available water molecules
during the expansion from ambient pressure into the first pressure stage due to the significant cooling. During this experiment,
no larger water clusters were detected likely due to the use of clean and dried air having a low relative humidity (RH) of
approximately 2 %. Other experiments at higher RH showed hydronium clusters up to 1000 m/z and higher. The impact of
larger hydronium ions on the dV scan can be discarded in this study.
The high number of collisions in the first pressure stage leads to a thermodynamic equilibrium distribution of hydronium
clusters. Consequently, a dV scan in the second pressure stage affects only the established hydronium cluster distribution
coming from the first pressure stage.

**3.2.3 Threshold binding energies**
The results from section 3.2.2 allow establishing an approximate threshold cluster binding energy for a fragment-free transfer
through the mass spectrometer as an example for the applied conditions. These results may vary under different conditions.
To estimate this threshold for the ioniAPi-TOF, we start with the $H_3O^+(H_2O)_3$ cluster ion. From the declustering scan in Fig.
7, the decrease of the ion signal of $H_3O^+(H_2O)_3$ starts at a voltage difference (dV) of -3 to -4 V. Below these dVs, fragmentation
is not a significant issue between the skimmer-1 and the second hexapole. Therefore, we conservatively estimate that cluster
ions with binding energies above 17 kcal/mol are likely to be transferred through the probed region of the ioniAPi without
substantial fragmentation for a low fragmenting setting. Cluster ions with binding energies below this threshold are partially
affected by fragmentation with increasing degree. Assuming a linear relationship between the voltage difference and the
binding energy according to Lopez-Hilfiker et al. (2016), we extrapolate a threshold binding energy of 8 to 11 kcal/mol using
the $dV_{50}$ values from Table 4 for the ion transfer between skimmer-1 and the second hexapole. Other regions were shown to
be less critical. Below this threshold, cluster ions are not likely to be detected depending on other conditions in the ioniAPi.
It has to be noted that the ion transmission shows strong responses for even small voltage differences between ion optic parts.
A DC offset of only 0.2 V on the second hexapole for example can significantly improve the ion transmission compared to no
offset. To maintain a satisfying detection sensitivity the electric potentials of the second hexapole and the following lens should
be set closely.

To compare the threshold binding energy of fragment-free cluster transfer of the ioniAPi-TOF to a quadrupole based APi-TOF, we only can give rough estimates based on existing literature. Via comparing modelled binding energies for adduct cluster ions and their sensitivity with a ToF-CIMS , Iyer et al. (2016) estimated that cluster ions with a binding energy below 25 kcal/mol can be expected to fragment at least partially during the ion transfer for the ToF-CIMS in Lopez-Hilfiker et al. (2016) and that cluster ions of binding energies below 10 kcal/mol are not likely to survive the transfer. Although it is not clear from their study if fragmentation can happen in the IMR or in the APi of the instrument, they conclude in the supplement that fragmentation in the APi is more likely (Iyer et al., 2016).

In Brophy and Farmer (2016), a declustering (dV) scan of the acetate-acetic acid cluster ($C_2H_3O_2^-(C_2H_4O_2)$) is shown. The voltage difference was also scanned between the skimmer and the front of the second multipole as done in this study (Fig. 4 in Brophy and Farmer). For this region, the authors determined a $dV_{50}$ of 4.1 V for $C_2H_3O_2^-(C_2H_4O_2)$ which has a binding energy of 29.3 kcal/mol (Meot-Ner and Sieck, 1986). At a voltage difference of 0 V, this cluster did not completely reach a plateau which must be considered as still partially fragmenting. From this, the threshold binding energy for their instrument seems to be even above the ToF-CIMS in Iyer et al. (2016). In contrast, Bertram et al. (2011) showed a mass spectrum of acetate-acetic acid cluster ions where under weak electric fields (15 V/cm throughout the APi) also higher order clusters ($C_2H_3O_2^-(C_2H_4O_2)_{1-2}$) were detectable with their ToF-CIMS instrument (Bertram et al., 2011). The binding energy of the trimer ($C_2H_3O_2^-(C_2H_4O_2)_2$) is 19.6 kcal/mol (Meot-Ner and Sieck, 1986). From Bertram et al. (2016), also a lower fragmenting transfer of cluster ions for a quadrupole based APi-TOF is possible. While no quantitative threshold binding energy was determined, it can only be estimated to be in the order of the binding energy of the acetate-acetic acid trimer of 19.6 kcal/mol. The differences in thresholds of fragment-free cluster transfer for the mentioned instruments depend obviously on more factors than the applied voltage settings in the APi like instrument geometry, pressures and flows. Nevertheless, our data suggests that the critical region of the ioniAPi is between the skimmer and the entrance of the second hexapole and that it allows a slightly lower threshold binding energy for the transfer of cluster ions. From our data, it is still difficult to attribute the observed difference to the number of poles of the ion guides. In the case of RF-only ion guides, this difference could be explained only by the radial contribution of the multipoles. Here, more research is needed regarding the effect of RF-frequency and amplitude on cluster ions at different pressures. For example, Rus et al. (2010) concluded that RF heating in the multipole was responsible for fragmentation of unstable cluster ions. Further, the successful fragmentation of a cluster via a collision with air as buffer gas depends also on the achieved collision energy in the centre-of-mass system. Heavier ions need higher electric fields to achieve the necessary collision energy. But they also can more readily accumulate the collision energy in a higher number of vibrational modes within the cluster compared to smaller ions reducing their chance of fragmentation (Zapadinsky et al., 2019).

### 3.3 Mass window and comparison to a quadrupole based APi-TOF MS

In the course of the CLOUD 12 campaign, we conducted an inter-comparison with the quadrupole based APi-TOF mass spectrometer of the University of Eastern Finland (UEF). The results of the transmission efficiency inter-comparison in

positive ion mode made at the end of the campaign are shown in Fig. 8. The data points are corrected for cluster fragments as
mentioned in chapter 3.1. Here, inlet line losses are not accounted for as the calibration setup of the CCU allows nearly identical
flow conditions for both detectors, electrometer and mass spectrometer. A transmission efficiency of overall about 1% was
found for both mass spectrometers. The overall ion transmission is a factor of 2 to 3 higher for the UEF APi-TOF. This factor
can be attributed to various differences in the instrument configurations as described in section 2.5, e.g. ion optic configuration,
geometry as well as flows due to pumping. It can be noted that due to the compact size of the TOF mass analyser of the
ioniAPi-TOF, it can be run at a higher duty cycle with an almost threefold higher extraction frequency. Due to the higher
extraction, more ions are detected leading to a comparable transmission efficiency with the UEF APi-TOF.
From Fig. 8, the UEF APi-TOF has a higher transmission for medium mass ions between 200 and 600 Th. At higher masses
at about 1000 Th, the difference in the transmission efficiency of both instruments decreases. This can be explained with the
different ion transfer properties of higher order multipoles as shown in Table 1. In general, hexapole ion guides allow a poorer
focusing compared to quadrupoles but are capable of transmitting a broader mass range. Examples below will demonstrate
these properties using parallel measurements.
A qualitative inter-comparison was performed during a CLOUD experiment at CERN where the ozonolysis of a mixture of α-
pinene and isoprene was studied at -50 °C. The experimental conditions for the inter-comparison of both APi-TOF instruments
are noted in Table 5. This experiment was chosen because oxidation of α-pinene is expected to form highly oxygenated
molecules (HOM) (Ehn et al., 2010; Kirkby et al., 2016) and therefore high mass ions. Another reason was the use of the
CERN π-beam, which yields increased ion concentrations inside the CLOUD chamber leading to higher ion count rates with
both APi-TOFs and a better signal to noise ratio (S/N). The mass spectra obtained by the ioniAPi-TOF and the UEF APi-TOF
are compared in Fig. 9. Ion count rates are corrected for diffusion losses with the Gormley-Kennedy equation (Bemgård et al.,
1996) for both instruments.
First, this inter-comparison shows that in general, the overall peak pattern for the experiment is comparable for both
instruments. Several "bands" consisting of combinations of C5- and C10 HOM appear in both mass spectra and show a similar
distribution, e.g. mass ranges 300 to 450 Th, 450 to 650 Th, 650 to 850 Th and 850 to 1050 Th. For example, the peaks at m/z
151, 153, 169 and 185 correspond to $C_{10}H_{15}O^+$, $C_{10}H_{17}O^+$, $C_{10}H_{17}O_2^+$ and $C_{10}H_{17}O_3^+$, respectively showing the same relative
intensity in the mass spectrum of the ioniAPi-TOF as well as in the APi-TOF. Further analysis of the mass spectral data is not
subject to the present study.
Second, comparing the peak intensities a difference in the dynamic range between both instruments, the UEF APi-TOF and
the UIBK ioniAPi-TOF, for ions above a m/z of roughly 350 can be seen. This can mainly be attributed to the differences in
mass resolution (for this experiment, ~5000 (APi-TOF) and ~1600 (ioniAPi-TOF)) leading to a higher dynamic range for the
UEF APi-TOF. Higher diffusion losses in the much longer inlet line during the experiment as well as differences in the ion
transmission efficiency (see Fig. 8) can additionally contribute to the lower dynamic range of the ioniAPi-TOF. The apparent
higher sensitivity of the ioniAPi-TOF for high mass ions can be explained with a higher background noise due to the lower

mass resolution. The correction for inlet line losses and the threefold higher extraction frequency, values given in Fig. 8, of this compact TOF mass analyser compared to the medium-sized APi-TOF contribute as well.

Third, the mass spectra show large differences for ions of masses below 100 Th. As the UEF APi-TOF is set to the high mass range setting, the high-pass mass filter property of the quadrupole leads to the low-mass cut-off disabling the detection of small ions in exchange for an increase in ion transfer and detection of high mass ions. The use of hexapoles as ion guides in the ioniAPi-TOF allows the detection of small ions below 100 Th and of high mass ions up to 1100 Th simultaneously as shown for the tested experimental conditions. Here, only the mass range up to 1100 Th is shown as ion count rates at higher m/z were too low in both instruments, setting a practical upper m/z limit for this comparison.

However, the calibration results shown in Fig. 8 suggest that both instruments have a comparable level of transmission efficiency for ions above 100 Th. From this perspective, the hexapole ion guides show beneficial properties when measuring a broad mass range. The loss of information on one end of the mass window, as evident here for the quadrupole system, is not necessary. We note that this effect is not exclusively limited to comparing hexapole with quadrupole systems, as progression to even higher order multipoles may further broaden the accessible mass range. However, this will be subject to a future study.

## 4 Conclusion

In the present study, we introduce an alternative type of Atmospheric Pressure-interface Time-Of-Flight mass spectrometer, the so-called ioniAPiTOF, with the main difference of using hexapoles as ion guides in the APi. We characterised the ioniAPi-TOF regarding ion transmission efficiency, mass range transmission and the effect of ion transfer properties on the cluster ion stability. We found that the overall ion transmission efficiency (so far tested from m/z 74 to 1640 Th) with hexapole ion guides is around 1 % and comparable to existing APi-technology using quadrupole ion guides. The detection limit for one-hour integration time is around $5 \cdot 10^{-4}$ ions/cm$^3$. The width of the transmitted mass range was found to be broader compared to a quadrupole based APi-TOF, when each instrument was using just one single setting. In atmospheric nucleation studies, this has the advantage of simultaneously detecting very small precursor ions, which can harbour information on nucleation precursor compounds, and the much heavier cluster ions that form during nucleation. Further, the effect of the ion transfer through the ioniAPi on the cluster stability and their fragmentation was studied. Using the system of $H_3O^+(H_2O)_n$ we were able to estimate that cluster ions with binding energies above 17 kcal/mol are not substantially fragmenting in the critical region between the skimmer and the second hexapole. From the literature, we estimated a threshold of roughly 20 to 25 kcal/mol for quadrupole based APi-TOF instruments. Comparing these numbers, a slightly less fragmenting ion transfer for the ioniAPi seems possible. Still, further work is needed to understand the differences in fragmentation inside various APi configurations and if the lower fragmenting transfer suggested for the ioniAPi is due to the number of poles or if other differences (e.g. pumping, geometry, voltage settings) are responsible. The mass resolution of ~2000 in the present study was limited by the use of a compact TOF mass analyser. Future focus lies on improving both mass resolution and the transmission efficiency.

*Data availability.* Data related to this article are available on request from the corresponding authors.

*Author contributions.* PM, SF, GS and ML did the measurements with the ioniAPi-TOF in Innsbruck while GS and ML
performed the measurements at the CLOUD experiment. ML did the data analysis of the ioniAPi-TOF experiments. AY
contributed with data obtained with the UEF API-TOF at the CLOUD experiment. ML wrote the manuscript and all authors
contributed to the final manuscript development.

*Supplement.* The supplement related to this article is available online at: https://www.atmos-meas-tech-discuss.net/amt-2019-
531  97/


*Acknowledgements.* We thank IONICON Analytik GmbH for providing an ioniTOF-1000 mass spectrometer and for the
support in the development of the ioniAPi-TOF MS. Furthermore, we thank CERN for support of the CLOUD experiment as
well as the CLOUD collaboration (www.cern.ch/cloud) for the opportunity to test the new instrument and for their support.
We thank the tofTools team for providing tools for mass spectrometry analysis.

*Funding.* This work is funded by the Austrian Science Fund, FWF (project no. P27295-N20), the Tiroler Wissenschaftsfonds
(nanoTOF-ICE), the University of Innsbruck promotion grant for young researchers, the Academy of Finland's Centre of
Excellence program (grant no. 307331) and the University of Eastern Finland Doctoral Program in Environmental Physics,
Health and Biology.

*Competing financial interests.* The authors declare the following competing financial interest: IONICON Analytik GmbH
plans to commercialise the ioniAPi-TOF.

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

| Ion guide properties | Quadrupole (2n=4) | Hexapole (2n=6) | Octopole (2n=8) | Higher order multipoles (2n>8) |
|---|---|---|---|---|
| Focusing power | High | Medium | Low | Lower |
| Field free region | Low | Medium | High | Higher |
| Mass range | Low | Medium | High | Higher |

**Table 1**: Qualitative comparison of the ion guide transfer properties of ideal multipoles with 2n poles (Gerlich, 1992).


| Name | Sum formula | Monomer $A^+(AB)_0$ m/z [Th] ($d_z$ [nm]) | Dimer $A^+(AB)_1$ m/z [Th] ($d_z$ [nm]) | Trimer $A^+(AB)_2$ m/z [Th] ($d_z$ [nm]) | Tetramer $A^+(AB)_3$ m/z [Th] ($d_z$ [nm]) |
|---|---|---|---|---|---|
| TMAI | $C_4H_{12}NI$ | 74.097 (1.05) | | | |
| TPrAI | $C_{12}H_{28}NI$ | 186.222 (1.16) | 499.349 (1.45) | 812.475 (1.66) | |
| TBAI | $C_{16}H_{36}NI$ | 242.285 (1.24) | 611.474 (1.55) | 980.663 (1.73) | |
| THABr | $C_{28}H_{60}NBr$ | 410.473 (1.47) | 899.863 (1.78) | 1389.254 (1.97) | |
| IL | $C_{15}H_{30}F_6N_2O_4S_2$ | 200.238 (1.15*) | 680.393 (1.5*) | 1160.703 (1.7*) | 1640.703 (1.9*) |

**Table 2.** Positive cluster ions, their corresponding mass-to-charge ratio m/z and the mobility diameter $d_z$ of Tetra-Methyl-Ammonium-Iodine
(TMAI), Tetra-Propyl-Ammonium-Iodine (TPrAI), Tetra-Butyl-Ammonium-Iodine (TBAI), Tetra-Heptyl-Ammonium-Bromide (THABr)
and Tributylmethylammonium-bis(trifluoromethylsulfonyl)imide (ionic liquid: IL) used in this work. A is the tetra-alkyl-ammonium part of
the neutral molecule, while B can be I or Br in the case of the first four compounds. *The mobility diameters for the ionic liquid were
determined in this study with an uncertainty of ±0.1 nm.

| | ioniAPi-TOF UIBK | UEF APi-TOF |
|---|---|---|
| Type of multipoles | Hexapole | Quadrupole |
| Multipole configuration | Straight and geometrically identical hexapoles | A short (SSQ) and a big (BSQ) segmented quadrupole |
| Diameter of critical orifice at MS entrance [mm] | 0.4 | 0.3 |
| Flow rate through orifice [L/min] | 1.1 | 0.8 |
| TOF-platform | ioniTOF1000, IONICON Analytik GmbH | H-TOF, Tofwerk AG |
| Mass resolution (FWHM) | 1500-2000 | ˜5000 |
| Extraction frequency [kHz] | 30 | 12.5 |

**Table 3:** Main technical differences of the ioniAPi-TOF UIBK compared to the UEF APi-TOF relevant for this study (Junninen et al.,
2010). Mass resolution and extraction frequency are setting dependent. Shown values were used during the CLOUD 12 campaign and
therefore valid for the inter-comparison in section 3.3.

| | $\Delta G$ (T=298K) | $-\Delta H$ | $dV_{50}$ (Fig. 7) | $dV_{50}$ (Fig. S5) |
|---|---|---|---|---|
| | [kcal/mol] | [kcal/mol] | [V] | [V] |
| $H_3O^+(H_2O)_1$ | -24.2 | 31.5 | | -6.2 |
| $.H_3O^+(H_2O)_2$ | -13.4 | 20 | -7.5 | -3.2 |
| $H_3O^+(H_2O)_3$ | -9 | 17 | -5.6 | -2.4 |

**Table 4:** Gibbs free energies, binding energies (BE($H_3O^+(H_2O)_{1-3}$) = $-\Delta H$) and corresponding $dV_{50}$ for $H_3O^+(H_2O)_{1-3}$ clusters (Meot-Ner, 1984) determined for a dV scan shown in Fig. 7 and in Fig. S5.

| run number | temperature | rel. humidity | ions | $NH_3$ | $SO_2$ | $O_3$ | $C_{10}H_{16}$ | $C_5H_8$ |
|---|---|---|---|---|---|---|---|---|
| | [K] | [%] | [cm$^{-3}$] | [ppb$_v$] | [ppb$_v$] | [ppb$_v$] | [ppb$_v$] | [ppb$_v$] |
| 1963.15 | 223 | 99 | ~2000 | 0 | 0 | 38 | 0.2 | 2.8 |

**Table 5:** Experimental conditions for the inter-comparison during run 1963.15 at the CLOUD experiment, CERN.





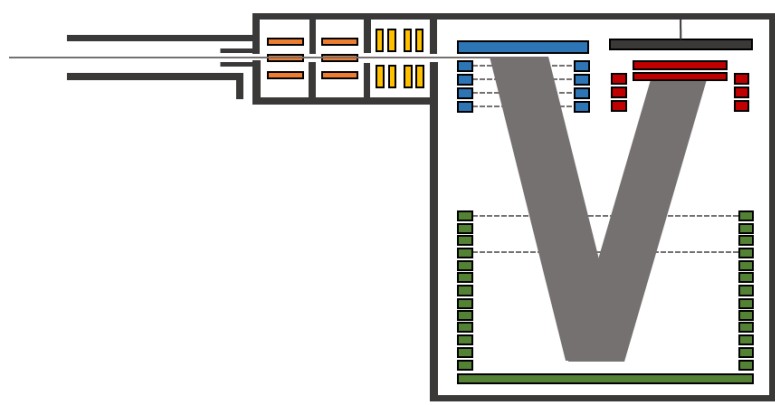


**Figure 1:** Schematic of the ioniAPi-TOF mass spectrometer. The hexapoles are shown in orange and the ion optical lens system in yellow.
The orthogonal extraction region is coloured in blue. The reflectron is coloured in green and the detection region with post acceleration
and MCP in red.

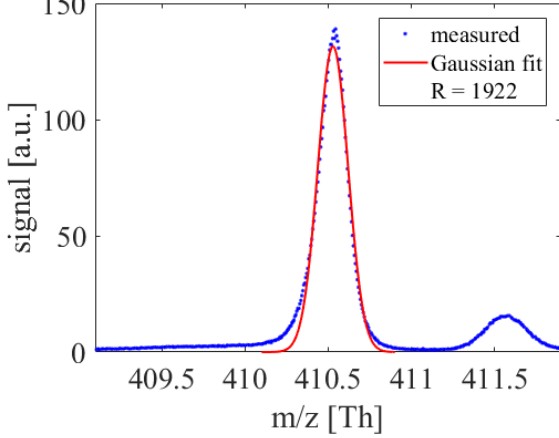


**Figure 2:** The mass resolution of the ioniAPi-TOF is about 2000 at a nominal mass of 410 Th, which corresponds to $C_{28}H_{60}N^+$, the
THABr monomer.

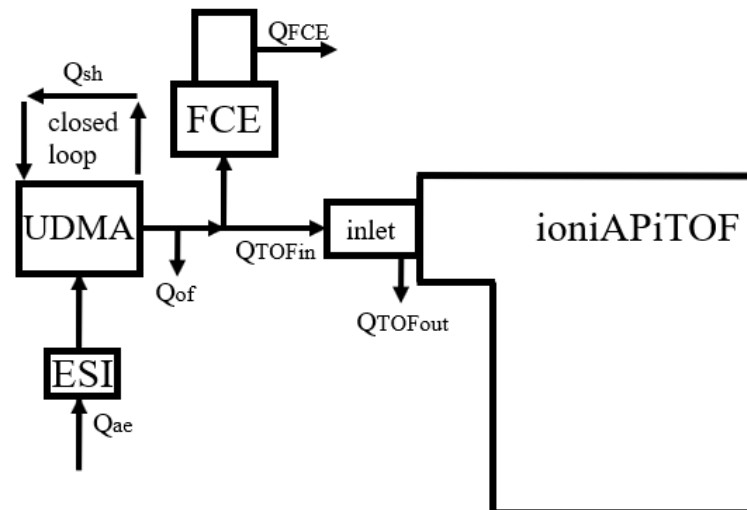


**Figure 3:** Experimental setup of the Cluster-Calibration Unit consisting of an electrospray ionisation source (ESI), a differential mobility
analyser (UDMA) and a Faraday cup electrometer (FCE) (Steiner et al., 2010; Winklmayr et al., 1991). Although not shown here, the flow
to both detectors is split via a Y-splitter with an angle of 20° for both sampling lines downstream to reduce inhomogeneity's that might
occur due to the flow separation.

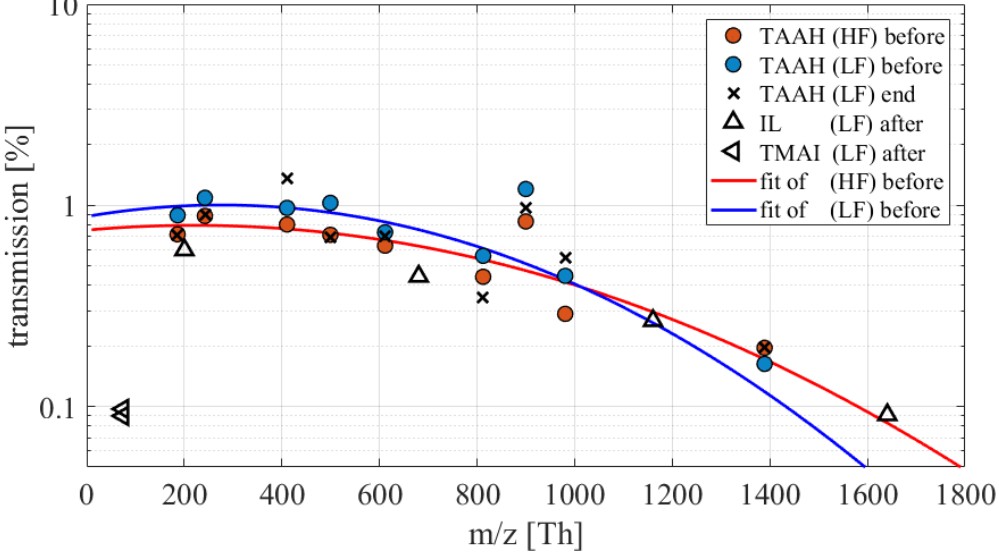


**Figure 4:** Transmission efficiency for low (LF) and high (HF) fragmenting ion transfer settings of the ioniAPi-TOF for ions of different
Tetra-Alkyl-Ammonium-Halides (TAAH) and an ionic liquid (IL), see Table 2. A Gaussian fit was used to obtain the transmission curves.
Calibrations were done before the CLOUD campaign, in the end and after the campaign in Innsbruck.

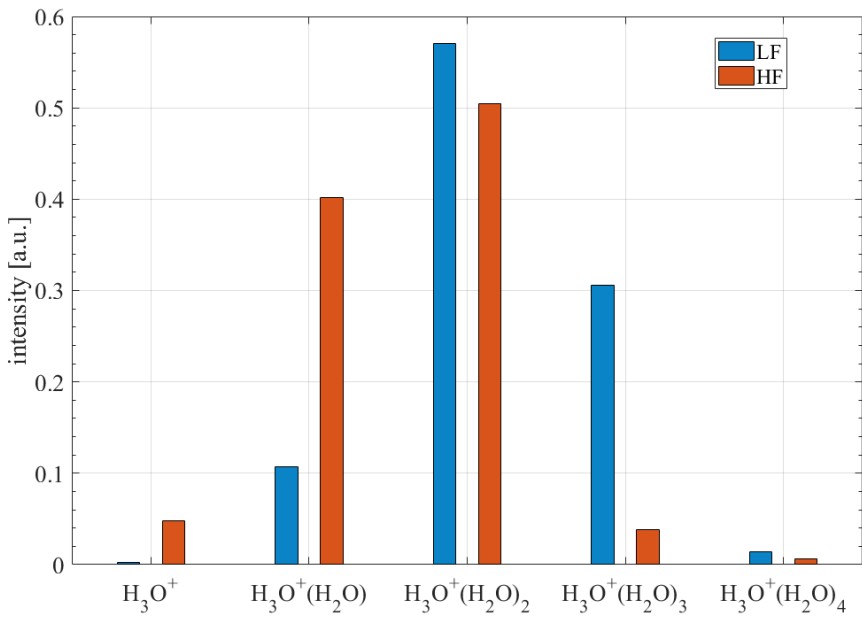

**Figure 5:** Comparison of the low fragmenting (LF, voltage difference: dV = -1.4 V) and the high fragmenting (HF, voltage difference:
dV = -10.0 V) setting. Ion counts are corrected for transmission effects and normalised for each setting.

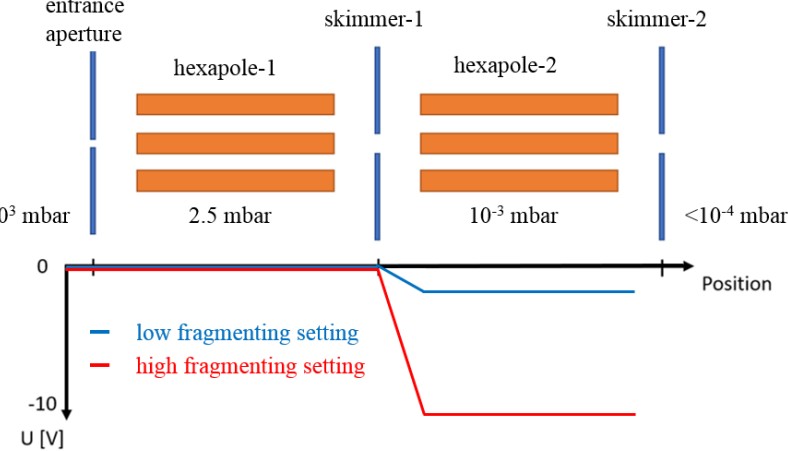

**Figure 6:** Schematic of the region inside the ioniAPi-TOF mass spectrometer where fragmentation was studied in this work. Here, a low
fragmenting clustered setting and a high fragmenting declustering setting can be used to identify cluster ions and to study their stability by
adjusting the voltage difference dV between skimmer-1 and hexapole-2.

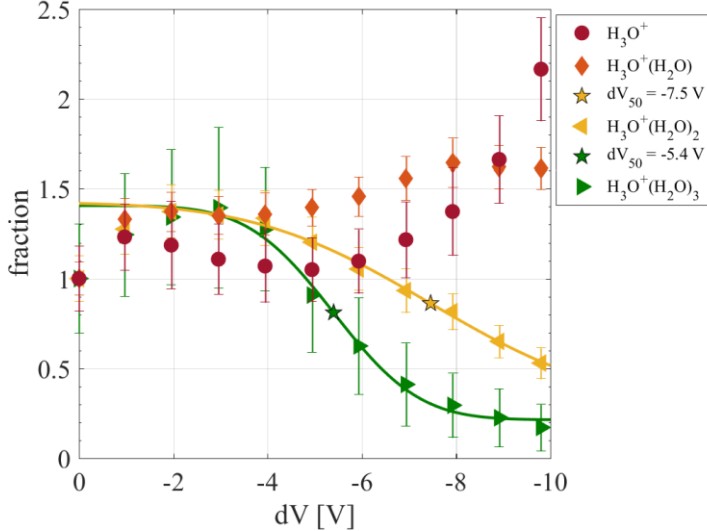

**Figure 7:** Declustering (dV) scan between the skimmer-1 and the second hexapole using hydrated hydronium clusters. Peak intensities are normalised on each ions' initial signal. The $dV_{50}$ of $H_3O^+(H_2O)_3$ is -5.4 V and the one of $H_3O^+(H_2O)_2$ is -7.5 V. The low fragmenting setting uses a dV of -1.4 V, whereas -10 V are used for the high fragmenting setting.



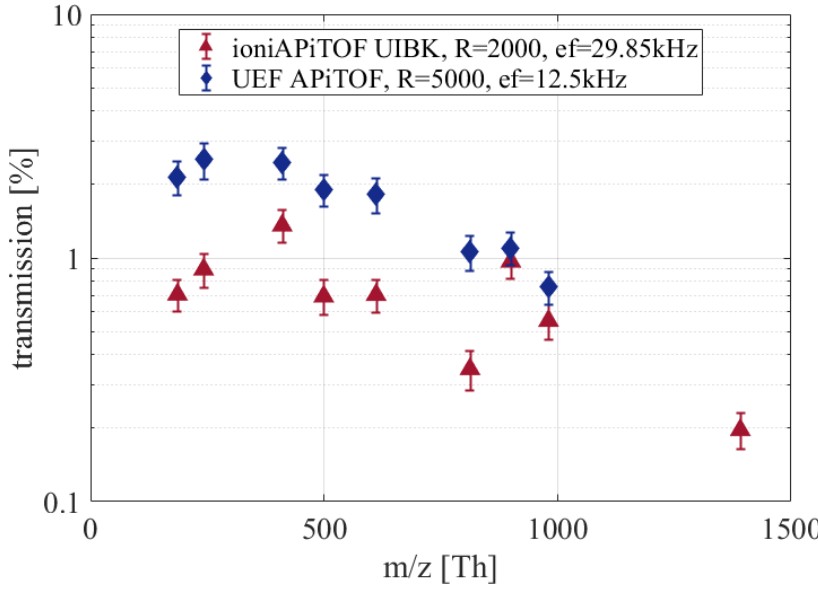


**Figure 8:** Comparison of the ioniAPi-TOF and the UEF APi-TOF regarding the transmission efficiency. The UEF API-TOF is set to high-mass range settings (m/z 100-2000 Th). The extraction frequencies *ef* varies with the length of the TOF mass analyser.

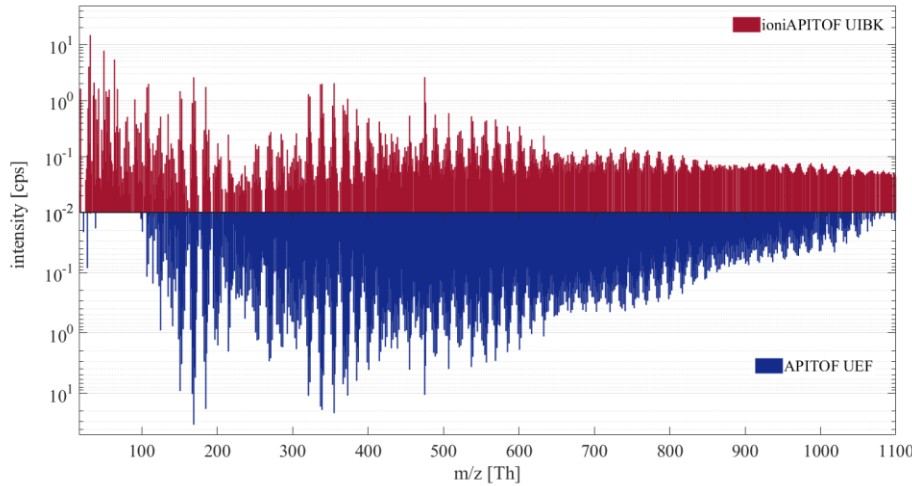


**Figure 9:** Comparison of the mass spectra obtained during run 1963.15 at the CLOUD experiment at CERN of the ioniAPi-TOF and the
UEF APi-TOF.