# Peer review of "Characterisation of the transfer of cluster ions through an Atmospheric Pressure interface Time-of-Flight mass spectrometer with hexapole ion guides"

_Atmospheric Measurement Techniques, 2019_

## Referee Comment (RC1) · Anonymous Referee #2 · 31 May 2019

This manuscript presents a new type of atmospheric pressure interface TOF mass spectrometer, the ioniAPi-TOF. The main difference to the earlier APi-TOF design is the change from quadrupole ion guides to hexapole ion guides. This is suggested to allow softer transmission of ions through the instrument, which may be possible, but is not convincingly shown in the manuscript, as detailed below. However, as so much of the current frontier research on both gas phase molecules and clusters have relied on solely the APi-TOF, it is a welcome addition to have a new instrument available for comparison. Overall the instrument is described in sufficient detail and I recommend

publication in AMT, following consideration of my comments and concerns below. In particular, the conclusion that the ioniAPi-TOF is able to transmit clusters bound by 17 kcal/mol needs to be more strongly shown.

Major comment:

The measurements of hydronium ions are used to deduce that the transmission of the ioniAPi is much less fragmenting than the APi-TOF. I see several issues with the interpretation of these results.

1. My main concern relates to the fact that no response at all is seen in the hydronium ion distribution before a dV of 4V. The authors seem to assume that this means that no cluster fragmentation is taking place at all. To me this rather seems to be a clear indication that this region of the ioniAPi is not the region where cluster fragmentation is mainly taking place (before the dV is increased high enough). If there is considerable fragmentation taking place somewhere after this region, with collisions able to break clusters of, say, 25kcal/mol, then it is not surprising that the dV in the probed region needs to go very high before starting to have an effect on the cluster distribution. The authors need to be more convincing with this data in order to be able to claim that this instrument is much less fragmenting than similar instruments.

2. As clusters are fragmenting, they will cause an increase in smaller water clusters. This is e.g. seen in the increase of the monomer and dimer clusters. There is no discussion about the potential influence of this on the trimer and tetramer when determining the dV50.

3. In Lopez-Hilfiker et al (2016), dV50 was defined as the voltage where the initial signal decreased to 50%. Here the voltage seems to be determined by the decrease halfway to the arbitrary maximum of the voltage scan.

4. The expansion from atmospheric pressure into the APi will cause significant cooling, and most likely formation of larger water clusters. This should be acknowledged and

considered potential impacts on the results.

Minor comments.

Line 16: "As we will show" is not commonly used in an abstract summarizing the work.

Line 32: "Owing" would read better as "having".

Line 74: Please clarify what "interactions with the reagent ion" means here.

Line 74: It is not customary to refer to the instrument being described here as a CI-APi-TOF. That term is typically used only for the atmospheric pressure "Eisele-type" CI inlet, while the one used here is generally called a ToF-CIMS.

Lines 87-100: The entire discussion around Eq. 1 is quite hard to follow, in part due to the complexity of multipole physics in general. I suggest the authors describe a bit more concretely what the function of V* is, and how it will affect the ions. Also, clarify the argumentation why the $(r/r_0)^4$ dependence causes a "larger field free region", as the simple interpretation of the equation is that the 4th power dependence of a hexapole would lead to a higher impact on the radial energy compared to the 2nd power dependence of a quadrupole.

Figure 3 is referenced before figure 2.

Lines 243-244: This HTOF lacking an APi needs some clarification, as sampling without any atmospheric pressure interface should not be possible here.
* * *

---

## Referee Comment (RC2) · Anonymous Referee #3 · 26 Jul 2019

This manuscript presents development of a similar instrument to the popular APi-TOF currently used in the measurements of atmospheric ions. The instrument uses two hexapoles instead of the traditional quadrupoles to guide the ions through the APi and shows some advances by using the hexapoles as the guides. Characterization of transmission efficiency, and ion transfer was presented. In addition, the mass windows between the IoniAPi-TOF and the APi-TOF were compared by concurrent measurements from the cloud chamber experiments. The manuscript is well organized and written, and certainly in the scope of the journal. However, some issues need to be

resolved before it can be publishable in the journal.

1. Why hexapoles not octapoles or higher order multipoles are employed in the instrument? What are the advantages by using the hexapoles as the ion guide compared to those high multipoles? There are also some literatures available in modeling ion trajectory for the multipoles. It will be of interest to the readers if the authors can add a brief literature overview in this aspect.

2. Figure 4 shows the overall transmission efficiency of the instrument. It would be beneficial to the readers if the authors provide detailed supplementary material information to illustrate the detailed process how to come up with the transmission efficiency curve since this is a very important part of the instrument calibration, e.g., what flow value was used, how to account for the fragmentation etc.

3. The distribution of the hydronium ion clusters with various voltage gradients between the skimmer and the second hexapoles shown in Fig. 7 is apparently dependent on many factors including the initial energy of the cluster (determined by the voltage applied in the entrance aperture), the voltage on the skimmer (currently grounded), the voltages on the lenses after the second hexapole. It is hence that the distribution may look differently under other settings. Did the authors tried any other settings? How those voltages may affect the distribution, especially how those voltages affect the detection sensitivity of the instrument?

4. In section 3.2.3, the authors assume that the binding energy is linear with the voltage gradient between the skimmer and the second hexapole, and estimate a threshold binding energy of about 8-10 kcal/mol for the hydronium ion clusters. The question is: how good is this assumption? It seems that the proposed binding energy is rather subjectable.

5. In Fig. 9, it is difficult to tell that the higher sensitivity from the ioniAPi-TOF in the higher mass range is real or just due to higher background noise since the signals are almost flat above 700 m/z for the instrument compared to the APi-TOF. In addition, the
comparison will be in a better view if the scale of the upper and bottom panels is set to the same physical height.

6. Some minor comments:

1) Some figures mentioned in the text were not in order. Please follow the correct order for the figure mention; 2) Consistency: for example, sometimes "at ground level" is used and "at ground levels"is used in other occasions; line 400, 406, 411 on p.13, First, Secondly, Thirdly. . ., please rephrase them to be consistent.

---

## Author Comment (AC1) · 22 Aug 2019

Published: 22.08.2019

We thank Referee #2 for his comments that will help us to improve our manuscript. In the following, we present our reply. The reviewer's comments (RC) are marked in *italic*, our replies (AC) follow in Times New Roman and the updated text is shown in blue.

*RC: 1. My main concern relates to the fact that no response at all is seen in the hydronium ion distribution before a dV of 4V. The authors seem to assume that this means that no cluster fragmentation is taking place at all. To me this rather seems to be a clear indication that this region of the ioniAPi is not the region where cluster fragmentation is mainly taking place (before the dV is increased high enough). If there is considerable fragmentation taking place somewhere after this region, with collisions able to break clusters of, say, 25kcal/mol, then it is not surprising that the dV in the probed region needs to go very high before starting to have an effect on the cluster distribution. The authors need to be more convincing with this data in order to be able to claim that this instrument is much less fragmenting than similar instruments.*

AC: We thank referee #2 for his comment on this topic. We agree that the way the data of Fig. 7 is acquired and illustrated led to the conclusion that no fragmentation is likely to happen before a dV of 4 V. There are two issues with Fig. 7. The first one is that a voltage of -13 V is applied during the dV scan to the lens (lens-1) that follows the second hexapole (hexapole-2). This was done to maintain the transmission during the scanning procedure. Unfortunately, this might have also been a source of fragmentation that could have dominated the first few steps of the dV scan. The second issue is that in Fig. 7 all signals are normalised to the sum of all four cluster ions. This was done to account for transmission effects during the scan. Consequently, the first few voltage steps appear flat in the response of the cluster ion distribution.

 We offer two solutions to these issues. First of all, we updated Fig. 7 by normalising the signals on each cluster ions' initial signal to follow the methods described in Lopez-Hilfiker et al. (2016).

In the updated version of Fig. 7, the response in the hydronium ion distribution is slightly different. Here, transmission effects for small dVs play a clearly visible role. With increasing dV, the hydronium ion distribution still shows only little changes. To address the referee's concerns, this could indeed be explained with fragmentation happening in another region of the ioniAPi. As we found the voltage of -13 V applied to lens-1 is the most likely reason. The new $dV_{50}$ values for the hydronium ions $H_3O^+(H_2O)_3$ and $H_3O^+(H_2O)_2$ show no dependence on the selected normalisation.

To understand such fragmentation in more detail and with reduced complexity, we further created a supplement and added a series of experiments.

Line 349: In the current configuration of the ioniAPi-TOF, no axial electric fields can be applied to any parts of the first pressure stage as explained previously. Therefore, the dV scan is obtained in a slightly different way compared to the one described in Lopez-Hilfiker et al. (2016). In Lopez-Hilfiker et al. (2016), the whole first pressure stage is shifted towards a more negative dV while the voltages downstream remain constant. In our case, no shift of the first pressure stage is currently possible. Therefore, the first pressure stage remains at ground potential.

Line 355: with a constant voltage of -13 V at the first lens that follows the second hexapole.

Line 382: As the voltage at the first lens is set to -13 V, fragmentation between the second hexapole and the following lens might dominate the first few voltage steps. To distinguish the role of the region from the skimmer to the entrance of the second hexapole and the region from the exit of the second hexapole to the following lens, we show additional experiments in the supplement. With a high-resolution ioniAPi-TOF, we conducted the same experiments and show with Fig. S4 that both instruments show nearly identical responses of the hydronium ion distribution to the dV scan procedure used in Fig. 7. In Fig. S5, a declustering scan between skimmer-1 and the second hexapole shows $dV_{50}$ values shifted to lower values, see Table 4 if the second hexapole and the following lens are stepped synchronously. In Fig. S6, a declustering scan between the exit of the second hexapole and the following first lens with lens 1 and lens 2 being stepped synchronously reveals that a voltage of below -9 V mainly increases the ion transmission. Only above -9 V the voltage difference from lens 1 to the exit of the second hexapole is high enough to induce fragmentation of the weaker bound hydronium cluster ions. Fig. S7 shows that a voltage scan between lens-1 an lens-2 has no effect on the hydronium cluster ion distribution. As a consequence, the region between skimmer-1 and the second hexapole is the region where voltage settings can be most effective on the fragmentation of cluster ions compared to the other probed regions.

Line 412: Below these dVs, fragmentation is not a significant issue between the skimmer-1 and the second hexapole.

Line 414: through the probed regions of the ioniAPi without substantial fragmentation for a low fragmenting setting.

Line 416: Assuming a linear relationship between the voltage difference and the binding energy according to

Lopez-Hilfiker et al. (2016), we extrapolate a threshold binding energy of 8 to 11 kcal/mol using the dV50 values from Table 4 for the ion transfer between skimmer-1 and the second hexapole. Other regions were shown to be less critical.

Line 420: It has to be noted that the ion transmission shows strong responses for even small voltage differences between ion optic parts. A DC offset of only 0.2 V on the second hexapole for example can significantly improve the ion transmission compared to no offset. To maintain a satisfying detection sensitivity the electric potentials of the second hexapole and the following lens should be set closely.

Line 443: Nevertheless, our data suggests that the critical region of the ioniAPi is between the skimmer and the entrance of the second hexapole and that it allows a slightly lower threshold binding energy for the transfer of cluster ions.

Line 513: Using the system of $H_3O^+(H_2O)_n$ we were able to estimate that cluster ions with binding energies above

17 kcal/mol are not substantially fragmenting in the critical region between the skimmer and the second hexapole.

*RC: 2. As clusters are fragmenting, they will cause an increase in smaller water clusters. This is e.g. seen in the*

*increase of the monomer and dimer clusters. There is no discussion about the potential influence of this on the*

*trimer and tetramer when determining the dV50.*

AC: We agree with referee #2. We added the following lines.

Line 374: $H_3O^+(H_2O)_2$ shows no significant response to the decrease of the $H_3O^+(H_2O)_3$ ion. This can be attributed to an overall low count rate of $H_3O^+(H_2O)_3$ and  a much higher count rate of $H_3O^+(H_2O)_2$. Fragmentation of

$H_3O^+(H_2O)_3$ will therefore not significantly increase the $H_3O^+(H_2O)_2$ count rate.

RC: *3. In Lopez-Hilfiker et al (2016), dV50 was defined as the voltage where the initial signal decreased to 50%.*

*Here the voltage seems to be determined by the decrease halfway to the arbitrary maximum of the voltage scan.*

AC: In Lopez-Hilfiker et al. (2016), the $dV_{50}$ was described as the voltage at half signal maximum. In most cases but not in all, the maximum agreed with the initial signal in that study. We updated Fig. 7 and normalised the signals of each cluster to its initial signal in agreement with Lopez-Hilfiker et al. (2016).

Line 360: The count rates of each ion are normalised to its initial count rate during the scan.

RC: *4. The expansion from atmospheric pressure into the APi will cause significant cooling, and most likely*

*formation of larger water clusters. This should be acknowledged and considered potential impacts on the results.*

AC:  We agree that this process is important to discuss. We added the following paragraph.

Line 398: A potential source of uncertainty on the experiments with hydronium cluster ions may be the fragmentation of larger hydrated hydronium clusters $H_3O^+(H_2O)_n$ with n>3. Such clusters could potentially form on collisions with available water molecules during the expansion from ambient pressure into the first pressure stage due to the significant cooling. During this experiment, no larger water clusters were detected likely due to the use of clean and dried air having a low relative humidity (RH) of approximately 2 %. Other experiments at higher

RH showed hydronium clusters up to 1000 m/z and higher. The impact of larger hydronium ions on the dV scan can be discarded in this study.

The high number of collisions in the first pressure stage leads to a thermodynamic equilibrium distribution of hydronium clusters. Consequently, a dV scan in the second pressure stage affects only the established hydronium cluster distribution coming from the first pressure stage.

*RC: Line 16: ”As we will show” is not commonly used in an abstract summarizing the work.*

AC: We thank Referee #2 for pointing this out.

In our case, hexapoles can accept and transmit a broad mass range enabling the study of small precursor ions and heavy cluster ions at the same time.

*RC: Line 32: “Owing” would read better as “having”.*

AC: We thank Referee #2 for this suggestion.

Typically, up to ten thousand ions per cm³ can be observed within the troposphere having a lifetime of a few hundred seconds.

*RC: Line 74: Please clarify what “interactions with the reagent ion” means here.*

AC: We rephrased line 74 accordingly.

It remains unclear, if this threshold can be explained by fragmentation in the APi or by the loss of weakly bound ligands during the charging process of a neutral cluster by the reagent ion in the Ion-Molecule-Region (IMR) of the ToF-CIMS (Kurten et al., 2014).

*RC: Line 74: It is not customary to refer to the instrument being described here as a CI-APi-TOF. That term is*

*typically used only for the atmospheric pressure "Eisele-type" CI inlet, while the one used here is generally called*

*a ToF-CIMS.*

AC: We correct the instruments' abbreviation in Lines 75, 76, 426, 427, 436, 438 to:

CI-APi-TOF -> ToF-CIMS

*RC: Lines 87-100: The entire discussion around Eq. 1 is quite hard to follow, in part due to the complexity of*

*multipole physics in general. I suggest the authors describe a bit more concretely what the function of V\* is, and*

*how it will affect the ions. Also, clarify the argumentation why the (r/r0)^4 dependence causes a "larger field free*

*region", as the simple interpretation of the equation is that the 4th power dependence of a hexapole would lead to*

*a higher impact on the radial energy compared to the 2nd power dependence of a quadrupole.*

AC: We improved the discussion around the effective potential as shown below:

Line 88: From theory, there are some differences with regard to the ion transfer properties comparing the quadrupole to higher order multipoles that can mainly be explained by the number of rods. To radially trap or guide ions of various mass-to-charge (m/z) ratios through a multipole a radiofrequency (RF) with amplitude $V_0$ is applied on alternating rods. Ions of low m/z are efficiently trapped with higher frequencies and lower amplitudes while ions of high m/z can be more efficiently transferred with lower frequencies and higher amplitudes. The time- averaged radial trapping field within a multipole of 2n electrodes can be described with the effective potential $V^*$

(Gerlich, 1992):

$$V^* = \frac{n^2}{4} \frac{q^2}{m\Omega^2} \frac{V_0^2}{r_0^2} \left(\frac{r}{r_0}\right)^{2n-2} \tag{1}$$

Here, we have the charge q, the ion mass m, the angular frequency $\Omega$, the amplitude $V_0$, the inner radius of the electrode arrangement $r_0$ and the radial distance of the ion r inside the multipole. In general, the effective potential

$V^*$ is high close to the rods and low close to the centre. The slope between multipole rods and its centre depends on the rod number, see Fig. S0. A higher rod number further provides a more homogenous trapping field. The trapping fields of RF-only multipoles do not affect the axial kinetic energy of ions, but can affect the radial ion energy (Armentrout, 2000).

From Equ. 1, it can be seen that the effective potential varies with $(r/r_0)^{2n-2}$. A quadrupole (n=2) has a quadratic dependence $(r/r_0)^2$ while a hexapole depends on $(r/r_0)^4$. Consequently, the effective potential of a quadrupole increases much closer to the centre of the ion guide compared to a hexapole. On the one hand, this results in an efficient focusing of the ions for a quadrupole, but on the other hand, this yields strong perturbations of ions in radial direction and thus, the ion kinetic energies are not well defined. Here, a hexapole has a much lower impact on the radial energy due to a larger field free region, as the effective potential is flatter close to the centre and higher close to the rods. Compared to higher order multipoles that have an even larger field free region, a hexapole still offers a more pronounced focusing power.

The $n^2$-dependence of the effective potential further means that for the same RF settings, a hexapole has a stronger trapping field over a quadrupole of a factor of 9/4. To transfer ions of high m/z with the same efficiency, a quadrupole would require higher RF settings which in turn would lead to an increased effective potential not only close to the rods but also in the centre according to the $(r/r_0)^2$ dependence of the effective potential.

*RC: Figure 3 is referenced before figure 2.*

AC: We thank Referee #2 for highlighting this. We changed the order of figure 3 and figure 2 in lines 184 and 169.

*RC: Lines 243-244: This HTOF lacking an APi needs some clarification, as sampling without any atmospheric*

*pressure interface should not be possible here.*

AC: We agree this requires more information. We added the following lines.

Lines 268: … with an H-TOF, Tofwerk AG Thun Switzerland, without an APi interface in the laboratory at the

University of Innsbruck (UIBK). This H-TOF was not equipped with a typical APi as it is part of a PTR-SRI-TOF

MS (Graus et al., 2010). For the experiments with the UDMA, we mounted a simple single pressure stage. This single pressure stage consisted of a front plate with a critical orifice diameter of 0.3 mm and two electrode lenses that were connected to the sampler plate of the H-TOF. A pressure of 2 mbar in the single pressure stage was achieved with a pre-pressure pump..

Additional references

Graus, M., Müller, M., and Hansel, A. (2010). High resolution PTR-TOF: quantification and formula confirmation of VOC in real time. J. Am. Soc. Mass Spectrom. 21, 1037–1044. doi: 10.1016/j.jasms.2010.02.006

Kurtén, T., Petäjä, T., Smith, J., Ortega, I. K., Sipilä, M., Junninen, H., Ehn, M., Vehkamäki, H., Mauldin, L.,

Worsnop, D. R., and Kulmala, M.: The effect of $H_2SO_4$ – amine clustering on chemical ionization mass spectrometry (CIMS) measurements of gas-phase sulfuric acid, Atmos. Chem. Phys., 11, 3007-3019, https://doi.org/10.5194/acp-11-3007-2011, 2011.

---

## Author Comment (AC2) · 22 Aug 2019

Published: 22.08.2019

We thank referee #3 for his comments. Those will help us to improve our manuscript. In the following, we present our reply. The reviewer's comments (RC) are marked in *italic*, our replies (AC) follow in Times New Roman and the updated text is shown in blue.

*RC: 1. Why hexapoles not octupoles or higher order multipoles are employed in the instrument? What are the advantages by using the hexapoles as the ion guide compared to those high multipoles? There are also some literatures available in modelling ion trajectory for the multipoles. It will be of interest to the readers if the authors can add a brief literature overview in this aspect*

AC: We thank referee #3 for his questions regarding our choice of the hexapole. In general, it would also be interesting to see if higher multipoles would yield even more advantageous properties for the transfer of weakly bound cluster ions. In this study, we chose the hexapole ion guide as the next logical step after the quadrupole because of its properties. These are increased mass range, lower effective potential close to the centre and an enhanced focusing compared to the octopole. The higher mass range of the hexapole might already be sufficient for the study of a broad range of atmospheric ions and clusters without a low-mass cut-off as known from the quadrupole. Other applications could of course profit from higher mass ranges or higher charge capacities using higher multipoles. We added a brief overview about ion trajectory simulations.

Line 127: Ion trajectory simulations through a quadrupole, a hexapole and an octopole by Hägg and Szabo (1986) showed that higher multipoles (n≥3) are more suited for guiding ions while only the quadrupole can be used as a mass analyser. In an accompanying study, the authors found that the transmission through higher multipoles depends on the initial conditions of the ion beam, e.g. initial position or velocity (Hägg and Szabo, 1986a). The reason for this is that the x- and y-coordinates are no longer independent compared to the quadrupole. An overall lower transmission efficiency of ions could be a likely consequence. Hägg and Szabo (1986) found in another study that the multipoles of even-order like the octopole have more stable trajectories because the opposing electrodes
have the same sign whereas multipoles of odd-order like the hexapole have opposing electrodes of opposite sign.
This would be one benefit using an octopole over a hexapole despite otherwise similar transfer properties. For
further ion trajectory simulations in multipole ion guides with focus on phenomena like collisional cooling and
radial stratification of different m/z ions due to ion-ion and ion-neutral interactions we refer the reader to
Tolmachev et al. (2003) and references therein.

*RC: 2. Figure 4 shows the overall transmission efficiency of the instrument. It would be beneficial to the readers if*
*the authors provide detailed supplementary material information to illustrate the detailed process how to come up*
*with the transmission efficiency curve since this is a very important part of the instrument calibration, e.g., what*
*flow value was used, how to account for the fragmentation etc.*
AC: We thank referee #3 for this suggestion. We created supplementary material information and added more
details about the way we determined the transmission efficiency.
Line 191: An exemplary ion mobility spectrum is shown in the supplement Fig. S1.
Line 197: For the same reason, the flow rates to both the FCE and the ioniAPi-TOF are set equally to 6 L/min
resulting in an overflow of 2 L/min.
Line 262: Additionally, minor peaks of impurities or fragments were observed, see Fig. S2.

*RC: 3. The distribution of the hydronium ion clusters with various voltage gradients between the skimmer and the*
*second hexapoles shown in Fig. 7 is apparently dependent on many factors including the initial energy of the cluster*
*(determined by the voltage applied in the entrance aperture), the voltage on the skimmer (currently grounded),*
*the voltages on the lenses after the second hexapole. It is hence that the distribution may look differently under*
*other settings. Did the authors try any other settings? How those voltages may affect the distribution, especially*
*how those voltages affect the detection sensitivity of the instrument?*
AC: We agree with referee #3 that the distribution may look differently under other settings e.g. m/z dependent
transmission effects by varying the RF settings of the first hexapole. In the current configuration of the ioniAPi-
TOF, no voltages can be applied in the entrance aperture besides the RF voltage on the first hexapole. Therefore,
no axial DC electric fields can have an impact on the tested hydronium cluster distribution. The voltage on the
lense after the second hexapole plays an important role regarding transmission effects and fragmentation. We tried
other settings and we provide the results of these experiments in the supplement.

Line 358: The initial cluster distribution may look differently depending on the conditions in the first pressure stage like pressure or electric fields, e.g. different RF settings on the first hexapole can alter the mass dependent transmission.

Line 420: It has to be noted that the ion transmission shows strong responses for even small voltage differences between ion optic parts. A DC offset of only 0.2 V on the second hexapole for example can significantly improve the ion transmission compared to no offset. To maintain a satisfying detection sensitivity the electric potentials of the second hexapole and the following lens should be set closely.

*RC: 4. In section 3.2.3, the authors assume that the binding energy is linear with the voltage gradient between the*

*skimmer and the second hexapole and estimate a threshold binding energy of about 8-10 kcal/mol for the*

*hydronium ion clusters. The question is: how good is this assumption? It seems that the proposed binding energy*

*is rather subjectable.*

AC: We agree with referee #3 that this threshold binding energy is only an estimation of the results from the dV

scanning procedure and not experimentally determined. Nevertheless, these values are given to present an orientation to the reader to classify the performance of the ioniAPi-TOF compared to one exemplary APi-TOF

from the literature.

Line 409: The results from section 3.2.2 allow establishing an approximate threshold cluster binding energy for a fragment-free transfer through the mass spectrometer as an example for the applied conditions. These results may vary under different conditions.

*RC: 5. In Fig. 9, it is difficult to tell that the higher sensitivity from the ioniAPi-TOF in the higher mass range is*

*real or just due to higher background noise since the signals are almost flat above 700 m/z for the instrument*

*compared to the APi-TOF. In addition, the comparison will be in a better view if the scale of the upper and bottom*

*panels is set to the same physical height.*

AC:  We thank referee #3 for his comment on Fig. 9. The apparent higher sensitivity of the ioniAPi-TOF is probably not that real. The lower mass resolution comes along with slightly higher background noise. In addition, the correction for inlet line losses yielded a higher correction factor as the inlet line of the ioniAPi-TOF was much longer. The scale could be changed to a more technical unit like ions per extraction. As we compare a medium sized TOF mass analyser (H-TOF, Tofwerk) and a compact TOF mass analyser (ioniTOF1000, Ionicon Analytik), differences in the analyser lengths will automatically result in different extraction frequencies. The extraction frequencies are given in Fig. 8 and show that a scaling on the extraction frequency will reduce the signals of the ioniAPi-TOF by a factor of roughly 3.

Line 488: The apparent higher sensitivity of the ioniAPi-TOF for high mass ions can be explained with a higher background noise due to the lower mass resolution. The correction for inlet line losses and the threefold higher extraction frequency, values given in Fig. 8, of this compact TOF mass analyser compared to the medium-sized

APi-TOF contribute as well.

*RC: 6. 1) Some figures mentioned in the text were not in order. Please follow the correct order for the figure*

*mention; 2) Consistency: for example, sometimes "at ground level" is used and "at ground levels" is used in other*

*occasions; line 400, 406, 411 on p.13, First, Secondly, Thirdly: : :, please rephrase them to be consistent.*

AC: 1) We corrected the order in lines 169 and 184. 2) We changed line 29 to "at ground level". 3) We rephrased all lines with "firstly, secondly, thirdly" to "first, second, third" for consistency.

Additional references

Hägg, C. and Szabo, I.: New ion-optical devices utilizing oscillatory electric fields. II. Stability of ion motion in a two-dimensional hexapole field, Int. J. Mass Spectrom. Ion Process., 73(3), 237–275, doi:10.1016/0168-

1176(86)80002-7, 1986.

Hägg, C. and Szabo, I.: New ion-optical devices utilizing oscillatory electric fields. IV. Computer simulations of the transport of an ion beam through an ideal quadrupole, hexapole, and octopole operating in the rf-only mode,

Int. J. Mass Spectrom. Ion Process., 73(3), 295–312, doi:10.1016/0168-1176(86)80004-0, 1986.

Hägg, C. and Szabo, I.: New ion-optical devices utilizing oscillatrory electric fields. III.stability of ion motion in two-dimensional octopole field, Int. J. Mass Spectrom. Ion Process., 13, 237–275, 1986.

Tolmachev, A. V., Udseth, H. R. and Smith, R. D.: Modeling the ion density distribution in collisional cooling RF

multipole ion guides, Int. J. Mass Spectrom., 222(1–3), 155–174, doi:10.1016/S1387-3806(02)00960-0, 2003.